# Gene therapy for progressive familial intrahepatic cholestasis type 3 in a clinically relevant mouse model

Nicholas D. Weber [1]*, Leticia Odriozola[2], Javier Martínez-García [2], Veronica Ferrer [3], Anne Douar [3], Bernard Bénichou [3], Gloria González-Aseguinolaza[1,2,4]* & Cristian Smerdou [2,4]*

Progressive familial intrahepatic cholestasis type 3 (PFIC3) is a rare monogenic disease caused by mutations in the *ABCB4* gene, resulting in a reduction in biliary phosphatidylcholine. Reduced biliary phosphatidylcholine cannot counteract the detergent effects of bile salts, leading to cholestasis, cholangitis, cirrhosis and ultimately liver failure. Here, we report results from treating two- or five-week-old *Abcb4*$^{-/-}$ mice with an AAV vector expressing human *ABCB4*, resulting in significant decreases of PFIC3 disease biomarkers. All male mice achieved a sustained therapeutic effect up through 12 weeks, but the effect was achieved in only 50% of females. However, two-week-old females receiving a second inoculation three weeks later maintained the therapeutic effect. Upon sacrifice, markers of PFIC3 disease such as, hepatosplenomegaly, biliary phosphatidylcholine and liver histology were significantly improved. Thus, AAV-mediated gene therapy successfully prevented PFIC3 symptoms in a clinically relevant mouse model, representing a step forward in improving potential therapy options for PFIC3 patients.

[1] Vivet Therapeutics S.L., Pamplona, Spain. [2] Division of Gene Therapy and Regulation of Gene Expression, Cima Universidad de Navarra, Pamplona, Spain. [3] Vivet Therapeutics S.A.S., Paris, France. [4] Instituto de Investigación Sanitaria de Navarra (IdISNA), Pamplona, Spain. *email: nweber@vivet-therapeutics.com; ggonzalez@vivet-therapeutics.com; csmerdou@unav.es

Progressive familial intrahepatic cholestasis (PFIC) describes a genetically heterogeneous group of inherited diseases, of which several types arise from abnormalities in a few transporters involved in bile production and regulation, which are mostly expressed in the canalicular membrane of hepatocytes[1]. In the case of PFIC type 3 (PFIC3, or MDR3 deficiency associated with PFIC), mutations affect the *ABCB4* gene, which encodes for multidrug resistance protein 3 (MDR3), a floppase involved in the translocation of phosphatidylcholine (PC) from the hepatocyte membrane to the bile. PC is an important component of bile, which is necessary to form mixed micelles with bile salts and cholesterol[2]. In the absence of PC, the bile acquires detergent properties and becomes very toxic to the membrane of hepatocytes and cholangiocytes, resulting in biliary canaliculi and epithelium injury, leading to cholestasis, cirrhosis, and ultimately liver failure[3]. In addition, the low level of phospholipids in bile often leads to crystallization of cholesterol, resulting in bile duct obstruction.

PFIC3 is a rare autosomal recessive disease that affects approximately 1 in 100,000 people worldwide[4]. More than 30 different mutations in *ABCB4* have been identified in PFIC3 patients, with one-third of mutations resulting in expression of a truncated MDR3 protein. Since mutations can alter the functionality of MDR3 in many different ways, the phenotype of PFIC3 patients shows a high degree of variability. Signs of cholestasis usually appear within the first year of life, although sometimes they are not detected until late childhood or even early adulthood[4]. In general, patients have high serum levels of bile salts and transaminases and moderate pruritus. The standard of care for PFIC3 patients is based on administration of ursodeoxycholic acid (UDCA), a hydrophilic bile salt that can replace more toxic hydrophobic bile salts in the liver[5]. However, only ~50% of patients benefit from UDCA therapy, and a liver transplant is the only curative treatment. Given the risks and high cost inherent with liver transplantation, new therapy options for PFIC3 patients are highly desirable. For example, gene therapy approaches based on gene correction or supplementation could provide a definitive cure for PFIC3 patients. Gene therapy is increasingly demonstrating its value as an innovative and safe therapeutic approach. Indeed, gene therapy based on the use of adeno-associated viral vectors (AAVs) has recently shown to provide safe long-term correction for several genetic diseases like haemophilia B[6,7] and Leber's congenital amaurosis[8].

A mouse model for PFIC3 in which the *Abcb4* gene was disrupted by elimination of its two first exons has been described[9]. These mice completely lack expression of MDR2 (the mouse ortholog for human MDR3), and are unable to secrete PC to the bile. This results in symptoms that reproduce most of the biomarkers and pathological signs of human PFIC3, including hepatosplenomegaly and liver fibrosis, although the severity and progression of the disease varies depending on the mouse strain[10].

In the present study, we have developed a therapeutic strategy for PFIC3 based on the administration of an AAV8 vector designed for liver-specific human MDR3 expression in a strain of *Abcb4*$^{-/-}$ mice that spontaneously progress to severe biliary fibrosis[11]. This therapy was able to increase biliary PC levels, resulting in normalization of all serum biomarkers, complete prevention of liver fibrosis and long-term correction of the disease in treated mice.

## Results

### Selection of the MDR3 variant
Since three potential isoforms have been identified for human MDR3 (A, B, and C)[12], we tested first the expression of each isoform in vitro. Isoform B differs from isoform A due to a seven-amino-acid insertion near the nucleotide binding domain 2, while isoform C has a 47-amino-acid deletion that includes transmembrane domain 11 (Fig. 1a). We generated AAV expression plasmids containing either the wildtype (wt) or a codon-optimized (co) version of human MDR3-A, -B, or -C cDNAs under the transcriptional control of the liver specific alpha-1 antitrypsin (A1AT) promoter (Fig. 1a). These plasmids were transiently transfected into the human hepatic cell line HuH-7 in order to analyse and compare expression and their capacity to transfer PC from the cell cytoplasm to the outer space of the cell (floppase activity) of the six variants. Utilizing immunofluorescence and confocal microscopy, cells that stained positive for MDR3 expression showed that only isoform A (wt and co) localized to the cell membrane (Fig. 1b). Meanwhile, isoforms B and C were expressed only in the cytoplasm, suggesting that these two variants are most likely non-functional at transporting PC across the cell membrane. When supernatant PC concentration from transfected cells was quantified, differences in floppase activity were detected, revealing MDR3 isoform A to be on average fourfold and 1.6-fold more active than isoforms B and C, respectively (Fig. 1c). There were no differences observed between the wt and co versions for any of the three isoforms.

### Liver expression via hydrodynamic injection of *Abcb4*$^{-/-}$ mice
To test for MDR3 expression in vivo, plasmids with the MDR3-Aco and -Awt variants, were tested in 7-week-old male *Abcb4*$^{-/-}$ mice via hydrodynamic injection. The animals were sacrificed 24 h later and MDR3 expression was analysed in liver sections by immunohistochemistry (IHC). The two mice inoculated with pAAV-MDR3-Aco showed clear MDR3 expression. In particular, one mouse showed MDR3 expression in discrete pockets throughout the liver tissue (Fig. 1d), with MDR3 clearly located on the canalicular membrane of hepatocytes. In contrast, pAAV-MDR3-Awt injection resulted in only one animal showing MDR3 expression that could be detected around a single cell from the entire sample (Fig. 1d). These results suggest that the MDR3-Aco sequence can be expressed in vivo much more efficiently than MDR3-Awt.

### AAV-MDR3-induced normalization of PFIC3 hepatic biomarkers
The *Abcb4*$^{-/-}$ FVB mouse model has shown to reliably replicate several PFIC3-associated markers, such as elevated serum liver transaminases, alkaline phosphatase (ALP), bile salts, and bilirubin; increased liver and spleen size; decreased concentration of PC in the bile; and severe morphological abnormalities in the liver, such as fibrosis, collagen deposits, and cell infiltrates[9]. The onset of symptoms in these mice appears very early, at least before 4 weeks of age[11].

We first tested the ability of AAV8 vectors harbouring the codon-optimized MDR3 isoform A cDNA (AAV-MDR3-Aco) to establish transgene expression in the livers of *Abcb4*$^{-/-}$ mice. Two-week-old mice treated via intravenous (IV) injection with AAV-MDR3-Aco at a dose of $1 \times 10^{14}$ viral genomes (VG)/kg showed robust hepatic MDR3 expression at 1 week after treatment. IHC staining of MDR3 showed diffuse expression throughout the liver, which was clearly localized to the biliary canaliculi (Fig. 2a, upper images) and constituted on average 62 ± 13% the level of MDR2 staining observed in *Abcb4*$^{+/+}$ (WT) animals. Twelve weeks after treatment, expression was still detected in the biliary canaliculi (Fig. 2a, lower images). However, expression levels had dropped to 31 ± 9.7% of WT mice in males and 8.5 ± 0.5% in females (Supplementary Fig. 1).

To test for a therapeutic effect of the AAV vector, we first treated 2-week-old male and female *Abcb4*$^{-/-}$ mice via IV

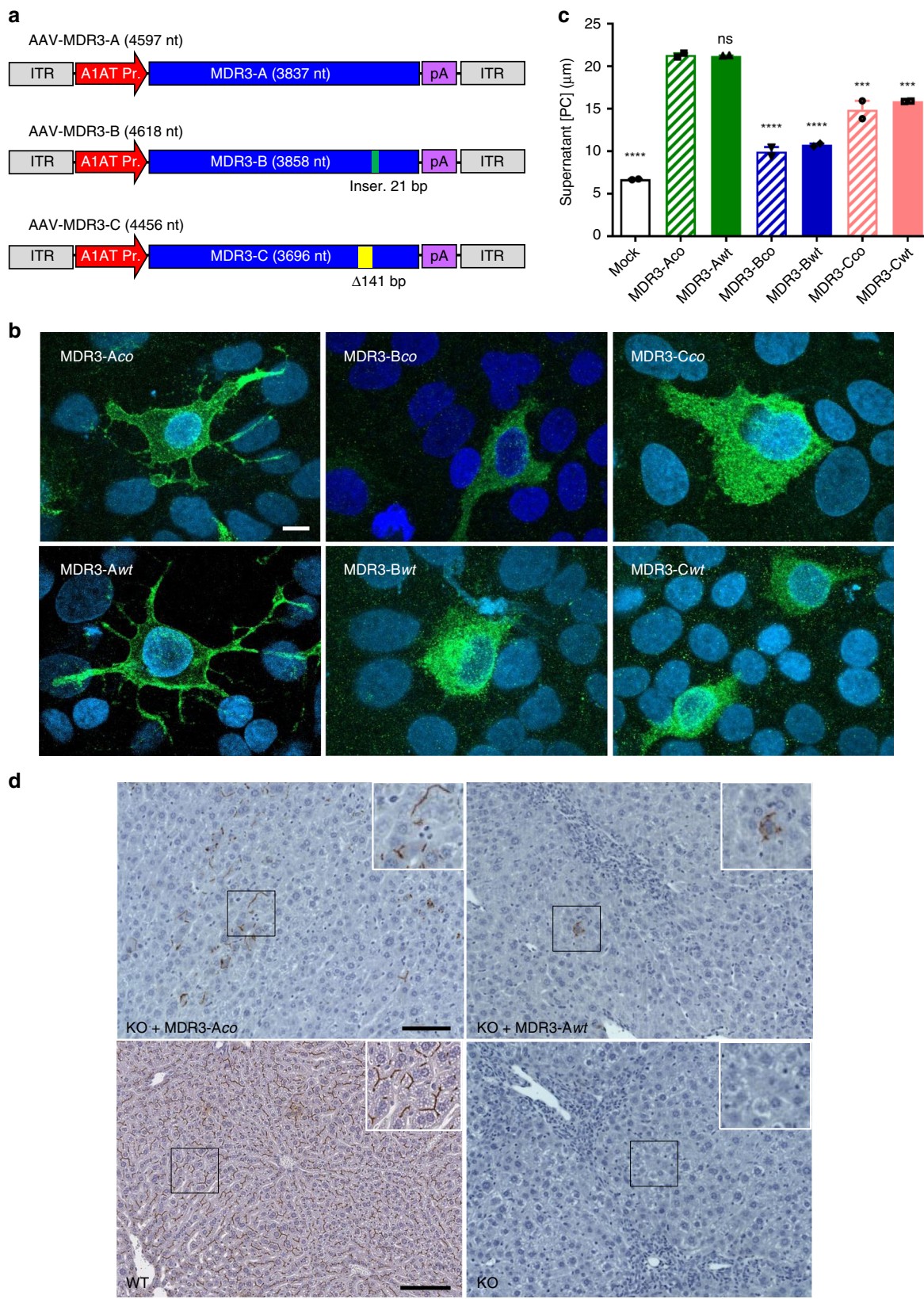

injection with AAV-MDR3-Aco at doses of $3 \times 10^{13}$ VG/kg (low) or $1 \times 10^{14}$ VG/kg (high) and monitored PFIC3 biomarkers in serum during the following 12 weeks. Male mice treated at the high dose achieved a sustained therapeutic effect up through 12 weeks, as ALP, alanine transaminase (ALT), aspartate transaminase (AST) and bile salt levels were all significantly reduced to levels similar to those observed in WT mice, used as controls. In contrast, neither saline-treated mice nor mice treated at the low AAV dose showed a significant reduction in these biomarkers (Fig. 3a). In females receiving the high dose, a

**Fig. 1 AAV-MDR3 vectors and analysis of plasmid expression. a** Diagrams of AAV vectors expressing MDR3 isoforms A–C downstream of the alpha-1 anti-trypsin promoter (A1AT Pr.). Insertion (Inser.) and deletion (Δ) present in isoforms B and C, respectively, are indicated. ITR, AAV inverted terminal repeats; pA, synthetic polyadenylation signal. **b** MDR3 expression in vitro. HuH-7 cells were transfected with AAV plasmids containing the indicated MDR3 isoform and analysed at 48 h by confocal immunofluorescence with an antibody specific against MDR3. For MDR3-Aco and MDR3-Awt, images correspond to representative serial planes of MDR3-positive cells showing clear membrane-localized expression. For isoforms MDR3-B and C, images showed a combination of all planes since no membrane staining was observed in any plane. Nuclei were stained with DAPI. Scale bar = 10 μm. **c** MDR3 activity in vitro. PC concentration in the supernatant of cells transfected with AAV plasmids was quantified by fluorometric assay. Cells transfected with a GFP-expressing plasmid served as negative control (mock). Equivalent transfection efficiencies were verified via qPCR with primers specific for A1AT promoter. Labels above individual bars indicate significance relative to MDR3-Aco (one-way ANOVA/Tukey's multiple comparisons test): ns, not significant, ***, $p < 0.001$; ****, $p < 0.0001$. Data are presented as mean ± standard deviation (SD). $F$ values and degrees of freedom (numerator, denominator): $F_{(6,7)} = 151.1$. Source data are provided as a Source Data file. **d** MDR3 expression in vivo. $Abcb4^{-/-}$ mice (KO) were HDI injected with AAV plasmids harbouring the indicated MDR3-A gene variant, and MDR3 expression was analysed by IHC with an anti-MDR3-specific antibody 24 h later. $Abcb4^{+/+}$ (WT) and non-treated KO mice were used as positive and negative controls, respectively. Representative pictures from one mouse in each group are shown. Scale bar = 100 μm.

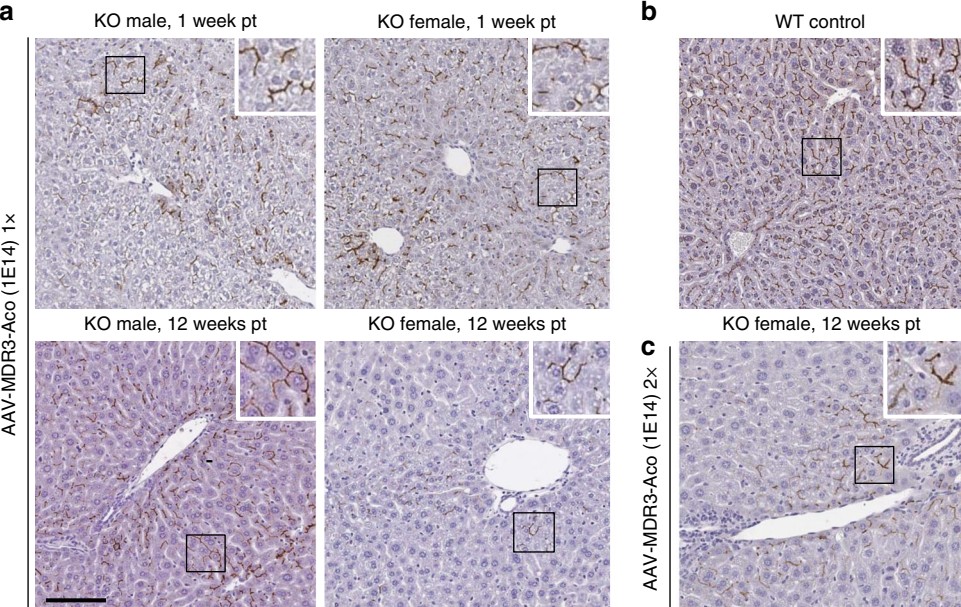

**Fig. 2 Analysis of AAV-MDR3-Aco expression in $Abcb4^{-/-}$ mice.** Two-week-old $Abcb4^{-/-}$ mice (KO) mice received one (**a**) or two (**c**) doses of $1 \times 10^{14}$ VG/kg of AAV-MDR3-Aco, and MDR3 expression was analysed at the indicated times after the first dose (pt) by IHC with an anti-MDR3-specific antibody. In **c**, the second dose was given when mice were 5 weeks old. A female WT mouse is shown as positive control for MDR2 staining (**b**). Representative pictures from one mouse in each group are shown. Scale bar = 100 μm.

therapeutic effect was also observed, which lasted 6 weeks in 100% of the animals. However, beginning at 8 weeks after treatment, half of the treated females experienced a waning of the therapeutic effect demonstrated with the rising of serum hepatic biomarker levels (Fig. 3b). The other half of the females maintained low biomarker levels up through 12 weeks.

In order to improve the efficacy of this therapy, we performed a second study in which $Abcb4^{-/-}$ mice were treated at 2 weeks of age with the AAV-MDR3-Aco high dose and received a second high dose 3 weeks later (at 5 weeks of age). In this case, $Abcb4^{-/-}$ males again showed values for serum hepatic biomarkers similar to WT mice (Supplementary Fig. 2), and $Abcb4^{-/-}$ females treated twice also showed a sustained therapeutic effect through the entire 12-week follow-up (Fig. 3c).

**Increase of bile PC and prevention of liver pathology.** We hypothesized that the improvement in serum PFIC3 biomarkers in $Abcb4^{-/-}$ mice treated with AAV-MDR3-Aco was most likely due to the restoration of PC secretion into bile, mediated by MDR3 expression in hepatocytes and localization to the cell

membrane of the biliary canaliculi. To evaluate whether this was the case, bile from treated mice was harvested from gallbladders upon sacrifice and PC concentration was measured. $Abcb4^{-/-}$ mice treated with AAV-MDR3-Aco at the high dose showed increased PC levels in the bile compared to saline-treated or low-dose-treated animals. Moreover, PC in the bile of animals treated with the high dose twice (2×) had on average two times higher PC levels than those treated only one time (1×) (Fig. 4a). However, the low vector dose did not have an effect on PC levels.

We also observed improvements in other PFIC3 symptoms. Liver and spleen sizes were decreased in the high-dose-treated animals (both 1× and 2×), as ratios of liver-to-body weight reached near WT values in the 2× females and in both 1× and 2× males (Fig. 4b and Supplementary Fig. 3). In addition to the much increased size and stiffness of the liver in the saline-treated $Abcb4^{-/-}$ mice, histological analysis revealed an advanced degree of liver damage, including fibrosis and cell infiltrates. As with the other disease markers, the degree of collagen staining was reduced significantly to near total elimination in high-dose AAV-treated $Abcb4^{-/-}$ mice in males (1× and 2×) and 2× females, indicating near complete loss of fibrosis (Fig. 4c, d). In 1× females, 50% of

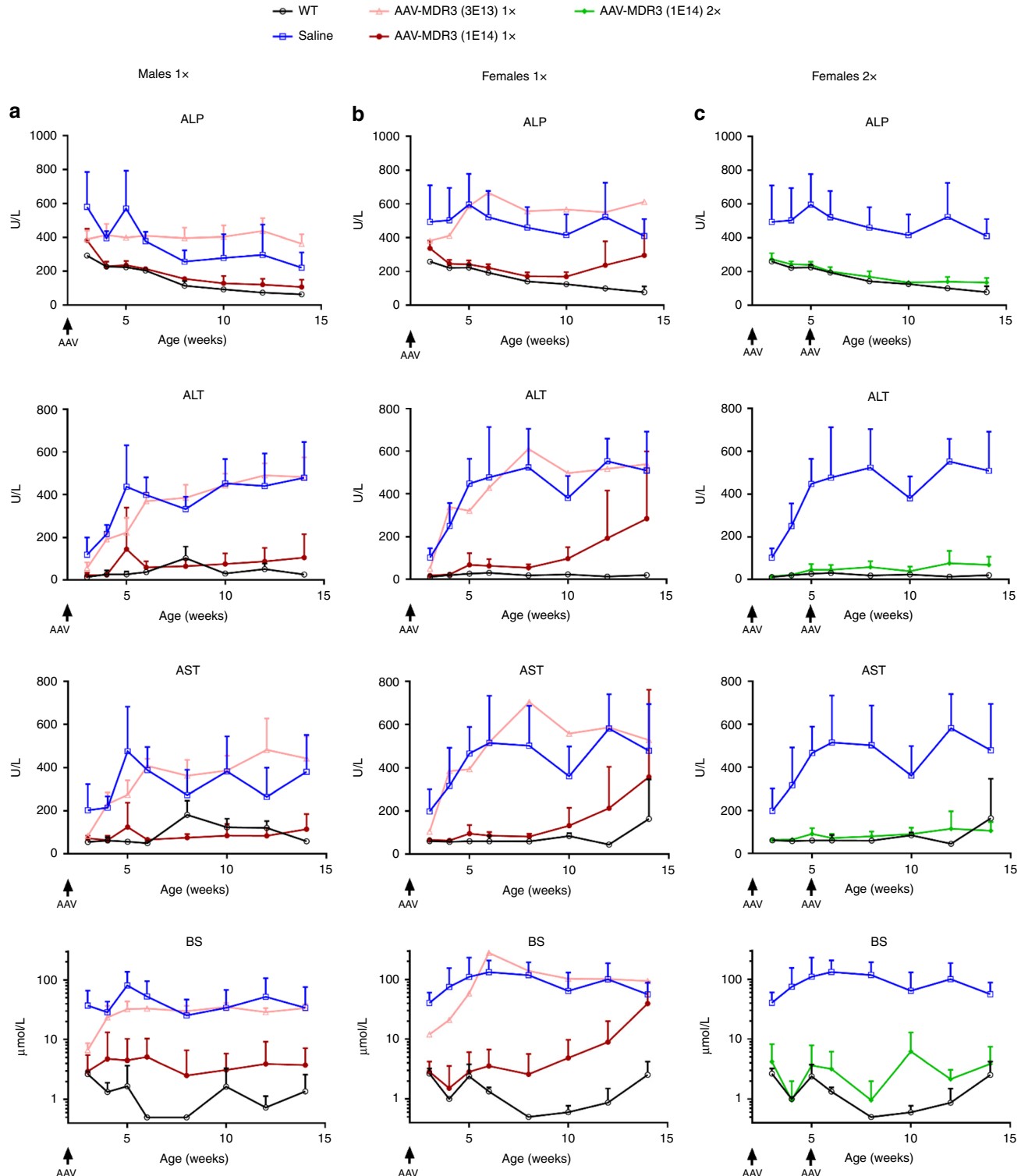

**Fig. 3 Serum biomarker levels for AAV-MDR3-Aco-treated *Abcb4*⁻/⁻ mice through 12 weeks.** The therapeutic effect of AAV treatment is shown in males (**a**) and females (**b** & **c**) that were treated with AAV-MDR3-Aco at $3 \times 10^{13}$ VG/kg one time (1×, pink, open triangles; $n = 3$ M (males)/1 F (females)), at $1 \times 10^{14}$ VG/kg one time (1×, red, filled circles; $n = 7$ M/6 F), or at $1 \times 10^{14}$ VG/kg two times (2×, green, filled diamonds; $n = 5$ F) using as controls saline-treated (blue, open squares; $n = 4$ M/8 F) and untreated *Abcb4*⁺/⁺ (WT) mice (black, open circles; $n = 3$ M/3 F). Data are presented as mean + SD. ALP alkaline phosphatase, ALT alanine transaminase, AST aspartate transaminase, BS bile salts. Source data are provided as a Source Data file.

animals showed a significant reduction of fibrosis, while in the rest, the degree of fibrosis was similar to the saline-treated mice (in Fig. 4d, a representative responder and non-responder female are each shown). The loss of fibrosis was the clearest evidence of the therapeutic effect of the treatment.

Although this strain of *Abcb4*⁻/⁻ mice has been reported to develop HCC, it takes 16 months for all mice to develop tumours[13]. Since our studies were not sufficiently long to evaluate this parameter, we analysed several markers of preneoplasia in our study population, which were sacrificed between 3.5 and

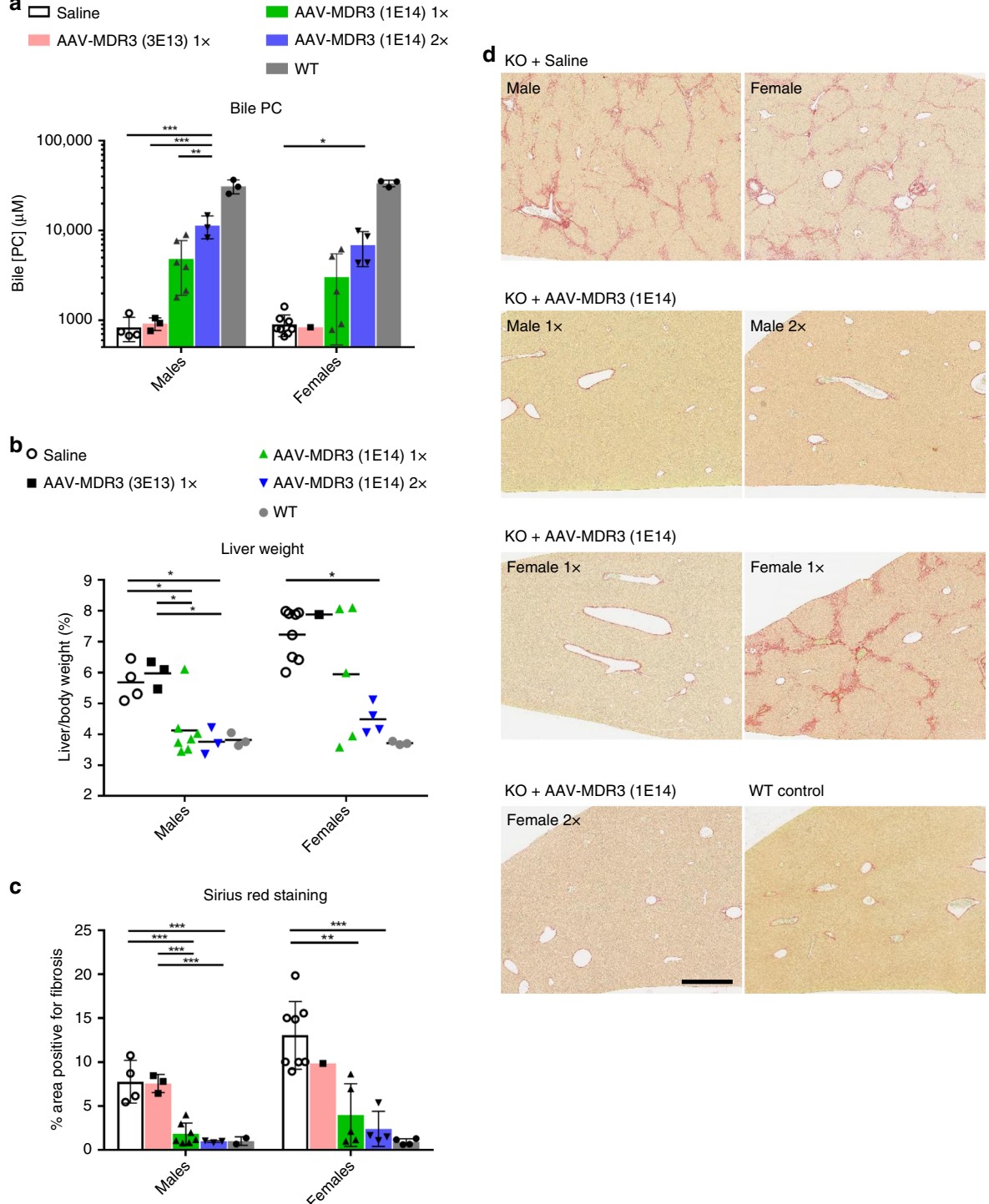

**Fig. 4 Biliary PC and evidence of cholestasis for *Abcb4*$^{-/-}$ mice treated with AAV-MDR3-Aco.** At the time of sacrifice, PC concentration in bile (**a**) and liver weight as a percent of body weight (**b**) were measured for *Abcb4*$^{-/-}$ mice treated with AAV-MDR3-Aco at $3 \times 10^{13}$ VG/kg 1× (pink bars or black squares; $n = 3$ M/1 F), at $1 \times 10^{14}$ VG/kg 1× (green bars or green triangles; $n = 7$ M/5 F), or at $1 \times 10^{14}$ VG/kg 2× (blue bars or blue inverted triangles; $n = 3$ M/4 F), as well as saline-treated controls (white bars or open circles; $n = 4$ M/8 F) and untreated WT mice (grey bars or grey circles; $n = 3$ M/3 F). Liver sections were stained with picrosirius red to indicate fibrosis (**d**) and the amount of tissue with red staining was quantified (**c**). Representative pictures from one mouse in each group are shown, except for females treated 1× at $1 \times 10^{14}$ VG/kg, for which one animal each from both responder (left image) and non-responder (right image) groups is shown. Scale bar = 500 μm. Statistics (one-way ANOVA/Tukey's multiple comparisons test): *, $p < 0.05$; **, $p < 0.01$; ***, $p < 0.001$; data are presented as mean ± SD; $F$ values and degrees of freedom (numerator, denominator): **a** males: $F_{(3,12)} = 14.52$, females: $F_{(3,15)} = 4.58$; **b** males: $F_{(3,13)} = 8.548$, females: $F_{(3,14)} = 4.501$; **c** males: $F_{(3,13)} = 23.13$, females: $F_{(3,14)} = 11.58$. Source data are provided as a Source Data file.

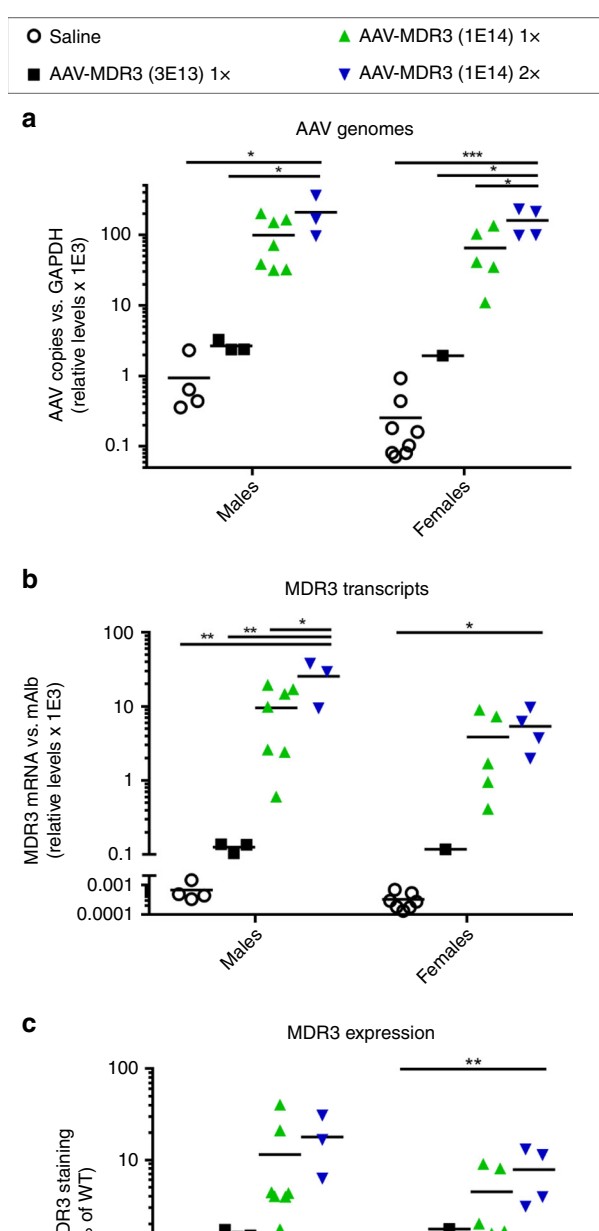

**Fig. 5 AAV transduction and transgene expression for $Abcb4^{-/-}$ mice treated with AAV-MDR3-Aco.** The levels of AAV genomes (**a**), MDR3 mRNA transcripts (**b**), and MDR3 protein expression (**c**) were quantified from liver tissue harvested from $Abcb4^{-/-}$ mice treated with AAV-MDR3-Aco at $3 \times 10^{13}$ VG/kg 1× (black squares), at $1 \times 10^{14}$ VG/kg 1× (green triangles), or at $1 \times 10^{14}$ VG/kg 2× (blue inverted triangles), as well as saline-treated controls (open circles). AAV genomes and MDR3 transcripts were quantified via qPCR and RT-qPCR, respectively, and protein expression was quantified as the percent area of tissue that stained positive for MDR3 expression and normalized to levels in WT mice. Animals were sacrificed between 12 and 16 weeks after treatment. Statistics (one-way ANOVA/Tukey's multiple comparisons test): *, $p < 0.05$; **, $p < 0.01$; ***, $p < 0.001$; data are presented as mean ± SD; $F$ values and degrees of freedom (numerator, denominator): **a** males: $F_{(3,13)} = 6.057$, females: $F_{(3,14)} = 12.74$; **b** males: $F_{(3,13)} = 7.59$, females: $F_{(3,13)} = 4.135$; **c** males: $F_{(3,13)} = 2.073$, females: $F_{(3,14)} = 5.696$. Source data are provided as a Source Data file.

trend towards improvement for GNMT and AFP (Supplementary Figs. 4, 5). For females, significant improvements were observed in AFP and γH2AX levels compared to either saline-treated or non-responder females. However, there was no significant difference between saline-treated $Abcb4^{-/-}$ and WT females for any of the markers, despite trends in all three markers. These results point to a likely reduction in tumour formation in mice that responded to our therapy.

**Transgene expression threshold needed for therapeutic effect.** DNA and RNA were separately extracted from liver tissue of the study animals upon sacrifice in order to quantify AAV-MDR3-Aco transduction efficiency and transgene expression. Vector transduction, as determined by AAV genome copies, was slightly better in males than females, and was 2- to 2.5-fold higher in the 2× mice compared to the 1× mice for both males and females (Fig. 5a). The level of MDR3 mRNA closely reflected the transduction efficiency (Fig. 5b). Of note, in the low-dose-treated mice, the relative levels of genomes and MDR3 mRNA were approximately 50- to 100-fold lower than in the 1× high-dose-treated mice, which could explain the lack of therapeutic effect observed in the low-dose-treated mice. There appears to be an inflection point in the dose response curve, below which transduction and/or expression are greatly reduced or lost over time, and above which a therapeutic effect is possible. In order to determine the inflection point with greater accuracy, we treated 2-week-old mice with additional intermediate doses and analysed vector genomes and MDR3 mRNA at time of sacrifice. This additional study showed that the inflection point is between $7 \times 10^{13}$ and $1 \times 10^{14}$ VG/kg (Supplementary Fig. 6).

IHC staining of liver sections for MDR3 protein confirmed that, as seen for transduction efficiency, MDR3 expression was higher in males than females. In addition, the 2× group showed about twice the amount of protein expression compared to the 1× group (Figs. 2 and 5c). Again, the low-dose-treated mice showed a markedly lower MDR3 expression, suggesting that loss of transgene expression might be more substantial in these animals. In order to determine whether the amount of MDR3 expression dictates the outcome of treatment, we analysed the correlation of MDR3 staining as measured by IHC and the severity of disease markers, including bile PC concentration, liver size, and fibrosis. For all markers of PFIC3, there was a clear correlation between the amount of MDR3 expression and reduction of disease marker severity (Supplementary Figs. 7–9). This analysis suggests that a threshold of around 2–3% of WT MDR3 expression would be sufficient to exert a therapeutic effect, as animals with expression

4.5 months of age. We tested expression levels of glycine *N*-methyltransferase (GNMT) and alpha fetoprotein (AFP) genes, which are downregulated and upregulated, respectively, in both human HCC and $Abcb4^{-/-}$ mouse liver tumours[14–16]. In addition, we also quantified the phosphorylated histone H2AX (γH2AX) by IHC, an indicator of DNA double-strand breaks that has been reported to be a good preneoplastic marker of HCC in $Abcb4^{-/-}$ mice[17]. Our results showed greater evidence of preneoplasia in saline-treated $Abcb4^{-/-}$ males than in saline-treated $Abcb4^{-/-}$ females compared to WT of the same age. In the males that responded to our AAV treatment (achieving bile PC concentrations above 3900 μM, see next section and Discussion), there was a significant improvement compared to either saline-treated or non-responder males for γH2AX and a

levels higher than this value demonstrated clear improvement in bile PC, liver size, and fibrosis severity.

**AAV-MDR3 is effective in 5-week-old *Abcb4⁻/⁻* mice.** We were interested in determining if our vector could induce a therapeutic effect in older mice, which already present liver inflammation and fibrosis (Supplementary Fig. 10), to see if an established PFIC3 disease phenotype is unconducive to AAV vector transduction. Therefore, we treated 5-week-old mice one time with $1 \times 10^{14}$ VG/kg of AAV-MDR3-Aco. Interestingly, treated mice achieved a clear therapeutic effect, as ALP, ALT, AST, and bile salt levels were all significantly reduced to WT levels, with the effect in males and half of the females lasting up through 12 weeks, while the effect waned in the other half of females starting about 4–5 weeks after treatment (Fig. 6). Improvements in biological parameters at sacrifice (hepatosplenomegaly, bile PC concentration, and fibrosis) were similar as in mice treated at 2 weeks of age (Fig. 7 and Supplementary Fig. 10). This supports the possibility of using our AAV vector in mice that already display a disease phenotype and that liver fibrosis might not constitute an exclusion criterion for potential candidates for therapy.

## Discussion

The possibility of addressing PFIC3 with gene therapy holds a great deal of promise, as it has several favourable aspects that make it highly suitable for developing this type of treatment. For instance, PFIC3 is a monogenic disease with the causative gene expressed exclusively in the liver, and its pathology includes very well-defined biomarkers[18]. To study the feasibility of PFIC3 gene therapy, we used a gene supplementation strategy based on an AAV8 vector expressing human MDR3 to transduce the liver of *Abcb4⁻/⁻* mice. The existence of three isoforms of MDR3[12] required a preliminary selection of the most suitable isoform to restore MDR3 expression and activity at the liver cell plasma membrane. This was performed by in vitro analysis, which clearly showed that isoform A was the only one of the three isoforms with membrane localization and thus with the potential for functionality in transporting PC into the bile. In vitro enzymatic activity analysis also pointed to the superiority of isoform A over isoforms B and C at exerting the specific phenotypic outcome, which reproduces findings from Siew et al.[19]. Further, a codon-optimized version was designed and evaluated in comparison to the wt version, as previous work has shown that the codon-optimized version of genes could lead to increased expression[20]. While we did not detect differences in expression levels or activity in vitro, we observed that the codon-optimized version was much more efficiently expressed than the wt version in the *Abcb4⁻/⁻* mice following plasmid delivery via HDI. While the reason for this discrepancy is not clear, there are some indications pointing to a greater instability of the MDR3-Awt cDNA sequence compared to MDR3-Aco. First, when cloning MDR3-Awt into AAV plasmids, frequent mutations and rearrangements were observed, which were eventually reduced by varying the bacterial strain and growing them at a lower temperature (30 °C). Second, production of the AAV-MDR3-Awt vector was very inefficient, resulting in titres that were routinely two logs lower than equivalent batches made with the codon-optimized version. These low titres, together with the low expression observed in the HDI experiments, pointed to the selection of AAV-MDR3-Aco as the best candidate vector.

In order to further evaluate AAV-MDR3-Aco for efficacy in correcting the disease, we used a clinically relevant model of PFIC3 based on *Abcb4⁻/⁻* mice with an FVB genetic background. Although this animal model presents more severe symptoms in females than males[10], the disease is sufficiently severe in either sex not to necessitate dietary challenge in contrast to a different

background strain that has been described by others[21]. We chose to treat 2-week-old *Abcb4⁻/⁻* mice because young mice represent a more suitable model for PFIC3 therapy since this disease manifests in humans early in childhood. In addition, older animals have a higher level of liver inflammation, a phenomenon that could potentially reduce AAV-mediated expression[22]. AAV transduction was similar in females and males, as observed in mice analysed 1 week after treatment (Fig. 2 and Supplementary Fig. 1). However, we observed a clear loss of MDR3 expression in both males and females when MDR3 staining was compared between 1 and 12 weeks after treatment (Supplementary Fig. 1), which was likely due to the still immature livers of these mice at the time of treatment (2 weeks), resulting in loss of AAV genomes in part through cell division due to liver growth[23]. Surprisingly, MDR3 expression at 12 weeks post-treatment was clearly lower in females compared to males (Figs. 2 and 5c). It has been observed by others that AAV-mediated gene transfer in liver is more efficient in males than in females[24,25] regardless of the promoter, transgene, or mouse strain, and was shown to be related to androgen-mediated stabilization of viral genomes in males[26]. In addition to the well-established increased transduction efficiency in males, the decreased level of expression in females after 12 weeks could be the result of the female livers being more diseased, which could lead to a higher loss of MDR3 expression over time due to liver cell regeneration. However, the loss of therapeutic effect was observed for only half of the females treated with a single dose and after 6 weeks. This loss of therapeutic effect could be the result of insufficient transduction in these animals, not allowing a minimal threshold of MDR3 expression to be reached to counteract the effects of the disease. In these conditions, the progression of the disease would therefore predominate, leading to further liver injury and regeneration, which, possibly in conjunction with other disease-related factors, could have caused a gradual loss of vector genomes, the subsequent decrease of MDR3 expression and drop in bile PC levels even further below the therapeutic threshold, resulting in bile toxicity, liver damage, and onset of severe cholestasis.

Evidence supporting the notion of a minimal expression threshold necessary to revert the disease has been reported already. De Vree et al. observed that PFIC3 disease could be reversed in the *Abcb4⁻/⁻* mouse model via transplanting these mice with primary hepatocytes from healthy WT mice[27]. With a repopulation of only 12% *Abcb4⁺/⁺* hepatocytes, a threshold was reached and the liver pathology was completely abrogated, with the correction still present after 1 year. We observed that mice treated at a lower AAV dose (30% of the high dose) showed no differences in disease symptoms from saline-treated animals. Quantification of MDR3-expressing cells for these livers showed expression to be below 2% of WT mice. Similarly, in the once-treated high-dose female group, half of the animals also had expression levels below 2% WT. These animals were those that lost the therapeutic effect 8 weeks after treatment and that upon sacrifice had the largest livers and spleens, lowest bile PC levels, highest degree of liver fibrosis, and greatest evidence of preneoplasia. We believe that this 2–3% WT level of MDR3-expressing cells constitutes a threshold above which enough PC is restored to the bile to prevent the development of PFIC3 symptoms. Based on our data, the threshold bile PC concentration above which the disease is prevented appears to be around 4000 μM (Supplementary Fig. 7). Since neither the exact amount of vector-mediated protein expression nor the activity of human MDR3 compared to mouse MDR2 is explicitly known, this threshold of bile PC concentration may be of more utility when translating this preclinical research to future studies. The threshold of 4000 μM of bile PC concentration represents about 12–13% the levels in our WT mice, which is comparable to the

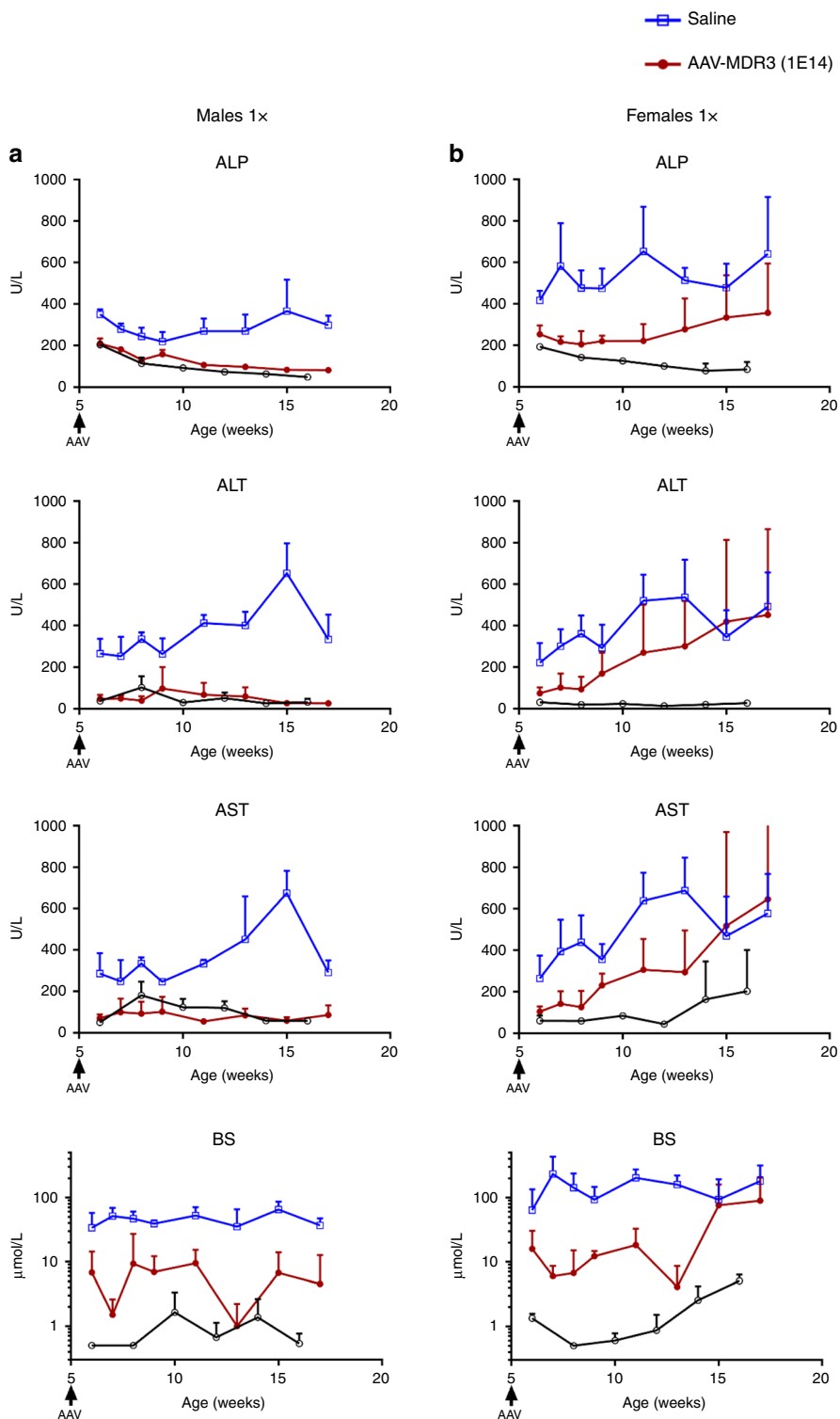

**Fig. 6 Serum biomarker levels for *Abcb4*⁻/⁻ mice treated at 5 weeks of age with AAV-MDR3-Aco.** The therapeutic effect of AAV treatment is shown in males (**a**) and females (**b**) that were treated with AAV-MDR3-Aco at $1 \times 10^{14}$ VG/kg one time (1×, red, filled circles; $n = 4$ M/4 F), using as controls saline-treated (blue, open squares; $n = 3$ M/4 F) and untreated *Abcb4*⁺/⁺ (WT) mice (black, open circles; $n = 3$ M/3 F). Data are presented as mean + SD. ALP alkaline phosphatase, ALT alanine transaminase, AST aspartate transaminase, BS bile salts. Source data are provided as a Source Data file.

amount of *Abcb4*⁺/⁺ hepatocyte repopulation reported by de Vree et al.[27]. In this way, our approach for PFIC3 could mimic gene therapies for diseases that involve secreted proteins[28], such as those for haemophilia A and B, α1-antitrypsin deficiency, and

Pompe disease[29], in which a small percentage of transduced cells may be sufficient to reverse the disease.

The *Abcb4*⁻/⁻ mouse model allowed us to carry out efficacy studies in very young mice, which may prove useful when

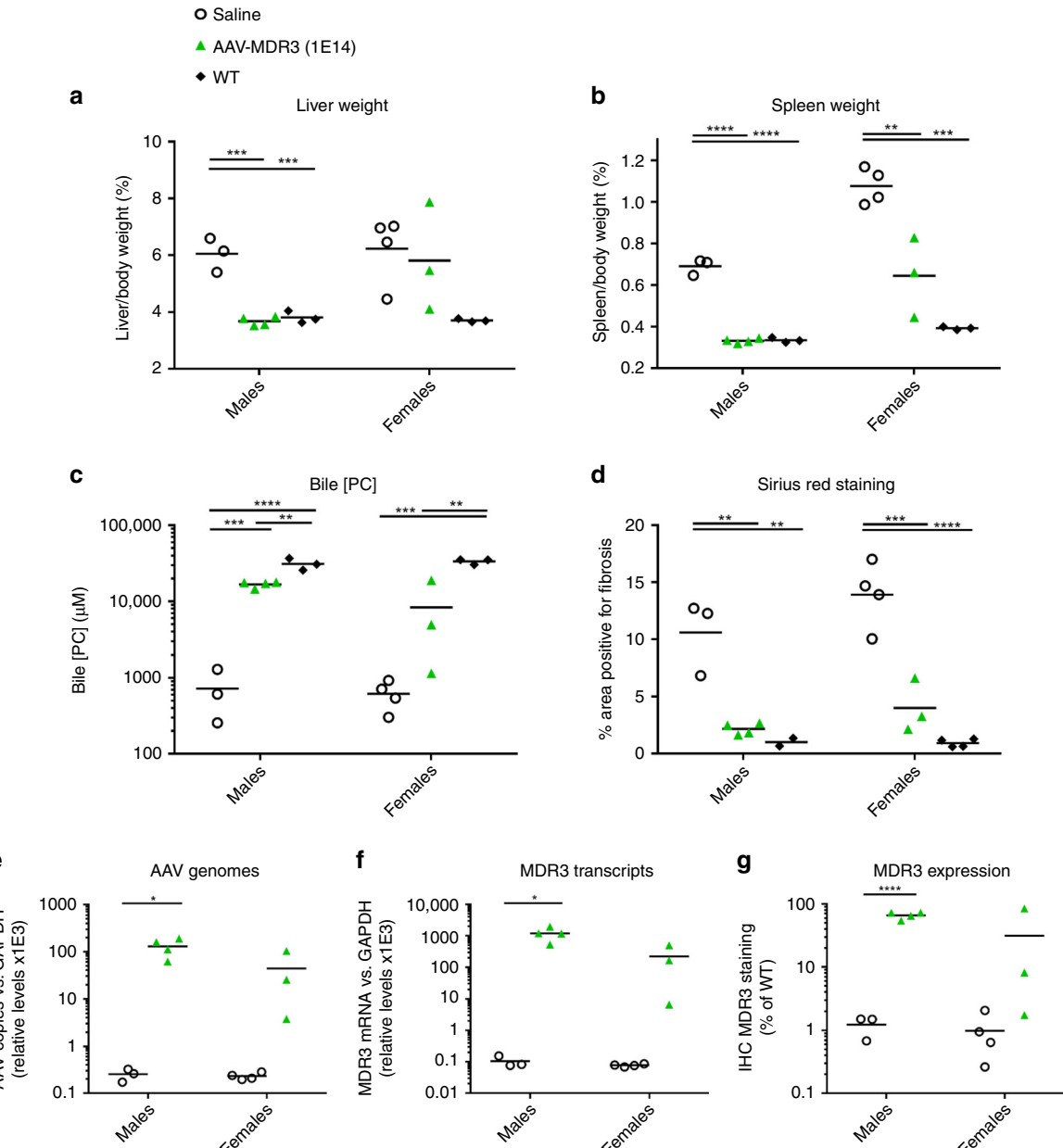

**Fig. 7 PFIC3 disease parameters in *Abcb4*⁻/⁻ mice treated at 5 weeks of age with AAV-MDR3-Aco.** At the time of sacrifice, liver (**a**) and spleen (**b**) weight as a percent of body weight, PC concentration in bile (**c**), liver fibrosis (**d**), AAV genomes (**e**), MDR3 transcripts (**f**), and MDR3 protein expression (**g**) were measured for *Abcb4*⁻/⁻ mice treated with AAV-MDR3-Aco at $1 \times 10^{14}$ VG/kg $1\times$ (green triangles; $n = 4$ M/3 F), and saline-treated controls (open circles; $n = 3$ M/4 F) and untreated WT mice (black diamonds; $n = 3$ M/3 F). Animals were sacrificed 12 weeks after treatment. Statistics (one-way ANOVA/Tukey's multiple comparisons test): *, $p < 0.05$; **, $p < 0.01$; ***, $p < 0.001$, ****, $p < 0.0001$. **a–d** F values and degrees of freedom (numerator, denominator): **a** males: $F_{(2,7)} = 44.52$, females: $F_{(2,7)} = 3.558$; **b** males: $F_{(2,7)} = 265.5$, females: $F_{(2,7)} = 30.81$; **c** males: $F_{(2,7)} = 69.93$, females: $F_{(2,7)} = 35.8$; **d** males: $F_{(2,6)} = 21.02$, females: $F_{(2,8)} = 39.88$. **e–g** Unpaired *t* test (*t*-value and degree of freedom): **e** males: ($t = 3.938$, df = 5), females: ($t = 1.737$ df = 5); **f** males: ($t = 3.557$, df = 5), females: ($t = 1.836$, df = 5); **g** males: ($t = 12.61$, df = 5), females: ($t = 1.37$, df = 5). Source data are provided as a Source Data file.

translating to the clinic, since children afflicted with PFIC3 will be a target population for treatment. Any variability in disease severity among individual animals may be a drawback of the model, but human PFIC3 patients also constitute a highly heterogenous population due to the myriad mutations in the *ABCB4* gene that lead to a wide range of phenotypes encompassing everything from a lack of MDR3 to enzymatic dysfunction to only impaired activity. It may turn out that achieving a therapeutic effect via gene therapy could be feasible in only a subset of the patient population, although we believe most if not all patients in

need of treatment could stand to benefit. Along the same lines, 50% of PFIC3 patients do not respond to the current standard of care with UDCA, with those that maintain a certain level of MDR3 activity being the ones that most benefit from this therapy[30,31]. If AAV-mediated gene restoration can increase MDR3 activity even only slightly in patients that do not currently benefit from UDCA, it may be sufficient to make them UDCA-therapy responders.

To our surprise, mice that were treated with an already established disease phenotype at 5 weeks of age also responded in

much the same way to the treatment as the 2-week-old mice. This bodes well for translation into the clinic where the patient population will already present symptoms of the disease at time of diagnosis.

We also report here an improvement in all parameters in mice treated twice with the AAV vector, which resulted in higher AAV transduction and MDR3 transgene expression compared to a single treatment, with no long-term adverse effects. The efficacy of the second treatment can be explained by immaturity of the immune system at the time of the first injection, or to the fact that the second dose was given prior to the peak anti-AAV neutralizing antibody level being reached. In the clinic, several options exist for developing therapies involving repeated administrations, including the use of different AAV serotypes for subsequent injections[32,33] and immune modulators to control the potential immune responses to AAV[34].

Two very recent studies published during the revision of this manuscript have also addressed the possibility of treating PFIC3 by using AAV-mediated gene therapy[19,21]. In one of these studies, therapeutic efficacy was only achieved by using a transposon-mediated integration strategy[19], something that could complicate translation into the clinic due to the potential oncogenicity of the treatment. In the second study, AAV-based therapy was tested in an $Abcb4^{-/-}$ mouse model with C57BL/6 background that does not develop PFIC3 symptoms unless challenged with a bile-acid-supplemented diet[21]. Although this therapy was successful, the absence of PFIC3 symptoms in these mice at the time of treatment greatly facilitates the transduction and maintenance of AAV vectors, a situation that will not be likely in patients.

In summary, we report here a sustained and significant reversal of PFIC3 disease biomarkers following treatment using an AAV vector harbouring a codon-optimized sequence of MDR3 isoform A in $Abcb4^{-/-}$ mice treated at 2 or 5 weeks of age. In a subset of the 2-week-old female population, a second AAV injection three weeks later was required to maintain the therapeutic effect. These results support the major interest of this approach as a curative therapy for this debilitating and life-threatening disease in patients who are faced with limited treatment options.

## Methods

**Cell lines.** HuH-7 (Japanese Collection of Research Bioresources Cell Bank: 0403) and HEK-293T (ATCC® CRL-3216™) cells were grown in DMEM (Gibco BRL) supplemented with 10% FBS, 2 mM glutamine, and 100 µg/mL streptomycin and 100 U/mL penicillin.

**Cloning and construction of AAV vectors.** Synthetic wildtype and codon-optimized sequences of human *ABCB4* cDNA coding for MDR3 isoforms A (NCBI Reference Sequence: NP_000434.1), B (NP_061337.1), and C (NP_061338.1) were obtained from GenScript (Nanjing, China) cloned into pUC57 flanked by SalI and NdeI sites (pU57-MDR3). To generate AAV-MDR3 plasmids, the DNA fragment coding for each MDR3 isoform variant was extracted by SalI and NdeI digestion from pUC57-MDR3 and subcloned into a pAAV2 cloning plasmid previously generated in our laboratory using the same restriction sites, resulting in pAAV-MDR3-Aco, pAAV-MDR3-Awt, pAAV-MDR3-Bco, pAAV-MDR3-Bwt, pAAV-MDR3-Cco, and pAAV-MDR3-Cwt plasmids. These plasmids contain wild-type AAV2 inverted terminal repeats (ITRs) and have the corresponding MDR3 open reading frame (ORF) downstream of the human A1AT liver-specific promoter[35] and a synthetic polyadenylation signal[36] at the 3′ end of the ORF.

**Production of AAV vectors.** To produce AAV viral particles (VPs) with serotype AAV8, 150-cm² flasks containing confluent HEK-293T cells were co-transfected using linear polyethyleneimine 25 kDa (Polysciences, Warrington, PA) with two different plasmids: pAAV-MDR3-Aco and pDP8.ape (Plasmid Factory, Germany), which contain adenoviral helper genes plus AAV2 rep and AAV8 cap genes. After 72 h, the supernatant was collected and treated with polyethylene glycol solution (PEG8000, 8% v/v final concentration) for 48–72 h at 4 °C. Supernatant was centrifuged at 1378×g for 15 min and the pellet was resuspended in lysis buffer (50 mM Tris-Cl, 150 mM NaCl, 2 mM MgCl₂, 0.1% Triton X-100) and kept at −80 °C. Cells containing AAV VPs were collected and treated with lysis buffer and

frozen at −80 °C. After three cycles of freezing and thawing, VPs obtained from cell supernatants and lysates were purified by ultracentrifugation at 350,000×g during 2.5 h in a 15–57% iodioxanol gradient[37]. Finally, the purified virus was concentrated using Amicon Ultra Centrifugal Filters-Ultracel 100K (Millipore). AAV-MDR3 vector titres (VG/mL) were determined by quantitative PCR (qPCR). VGs were extracted from DNAase-treated VPs using the High Pure Viral Nucleic Acid Kit (Roche). The primers specific for A1AT promoter were used for qPCR are listed in Supplementary Table 1. Vector titres ranged from $1 \times 10^{13}$ to $3 \times 10^{13}$ VG/mL.

**Analysis of MDR3 expression and activity in vitro.** HuH-7 cells were transfected with pAAV-MDR3 plasmids using lipofectamine 2000 (Thermo Fisher). Cells were fixed at 48–72 h and MDR3 expression was detected by immunofluorescence using a primary mouse monoclonal antibody specific for MDR3 (Millipore clone P₃-II-26, 1:100). A donkey anti-mouse Alexa-488-conjugated antiserum (Invitrogen ref. A21202, 1:1000) was used for detection. For activity analysis, after transfection cells were washed with fresh medium and incubated with DMEM containing 0.1% bovine serum albumin (BSA) in the presence of 0.5 mM sodium taurocholate (NaTC). After incubation for 24 h, the content of PC in the medium was determined using a Phosphatidylcholine Assay Kit (Sigma, MO) according to the manufacturer's instructions. Transfection efficiency was checked by using iQ™ SYBR Green (BioRad) in a CFX96 Real-Time Detection System (BioRad) with primers specific for the A1AT liver promoter that are listed in Supplementary Table 1.

**Animal studies.** FVB.Mdr2⁻/⁻ mice (FVB.129P2-Abcb4^tm1Bor) (*Abcb4⁻/⁻*) (JAX stock #002539)[9] as well as FVB.Mdr2⁺/⁺ (*Abcb4⁺/⁺*) mice (The Jackson Laboratory, Bar Harbor, ME) were bred in our own facilities (Cima Universidad de Navarra, Pamplona, Spain). Mice were housed with a 12 h light–dark cycle and received water and food ad libitum using a standard chow diet. In the case of plasmid-treated animals, mice were injected with 25 µg of pAAV-MDR3-Aco or pAAV-MDR3-Awt using the hydrodynamic injection-based procedure[38] and sacrificed 24 h later. Treatment with AAV vectors were performed in male and female mice at 2 and 5 weeks of age by retro-orbital IV injection. For sera analysis, mice were bled retro-orbitally at the indicated times. Liver, spleen, and gallbladder samples were collected from euthanized mice for organ size, PC determination, histological analysis, and nucleic acid extraction. All animal experiments and procedures were conducted in compliance to ethical regulations for animal testing and the studies were reviewed and approved by the Universidad de Navarra Institutional Ethical Committee (protocol numbers: 082c-17 for breeding and 086-17 for animal studies).

**Analysis of serum and bile samples.** Serum was separated from whole blood by centrifuging at 2300×g for 15 min in a microfuge. Serum ALP, ALT, AST, and total bile salts were quantified using a HITACHI C311 analyzer. For analysis of bile, mice were fasted for 24 h, sacrificed, and the gallbladder was carefully excised and placed in a 1.5-mL Eppendorf tube. Then the gallbladder was punctured with a 29G U100 insulin needle repeatedly to release bile, from which 2 µL were used to measure PC levels using the Phosphatidylcholine Assay Kit (Sigma) according to the manufacturer's instructions.

**Nucleic acid extraction and qPCR.** Vector genome copies present in liver extracts were determined by qPCR using iQ™ SYBR® Green (BioRad) in a CFX96 Real-Time Detection System (BioRad) with primers specific for the A1AT liver promoter. Mouse GAPDH (glyceraldehyde-3-phosphate dehydrogenase) was used as a normalizing gene (primers listed in Supplementary Table 1).

To analyse transgene and preneoplasia marker expression, total RNA was isolated from livers using the Maxwell® 16 LEV simplyRNA Tissue Kit (Promega) according to manufacturer's instructions and quantified. Extracted RNA was reverse transcribed into complementary DNA (cDNA) using M-MLV reverse-transcriptase (Invitrogen). qPCR was performed using TaqMan primers and probes specific for codon-optimized-MDR3 (FAM, NFQ-MGB) and mouse GADPH (VIC, NFQ-MGB) or mouse albumin (FAM, NFQ-MGB) designed by Applied Biosystems (sequences of which were not provided to the researchers) and run on a ViiA 7 (ThermoFisher). For neoplasia markers, qPCR was performed with iQ™ SYBR Green (BioRad) in a CFX96 Real-Time Detection System (BioRad) (primers listed in Supplementary Table 1) using GAPDH as a normalizing gene.

**Histological analysis of paraffin embedded sections.** Livers were collected from euthanized mice at the indicated time points. Formalin-fixed paraffin-embedded liver sections (3 µm thick) were dewaxed and hydrated. Antigen retrieval was performed using a solution of 20 µg/mL proteinase K for 30 min at 37 °C. For MDR3 staining, endogenous peroxidase was blocked with 3% H₂O₂ in deionized water and liver sections were incubated overnight at 4 °C with anti-MDR3 antibody diluted 1:2000 (LS-B5729, LS-Bio, Seattle WA) or with anti-phosphorylated histone H2AX (γH2AX) diluted 1:400 (#9718, Cell Signalling, Danvers, MA). After rinsing in TBS-Tween, samples were incubated with undiluted goat anti-rabbit EnVision (K4002, Dako, Glostrup, Denmark) for 30 min and peroxidase activity was revealed using 3,3′-diaminobenzidine (Dako). Finally, sections were lightly counterstained

with Harris haematoxylin, dehydrated, and coverslipped. Liver sections were also stained with haematoxylin and eosin (H&E) for routine histology and with picrosirius red to analyse liver fibrosis.

**Image analysis**. Several FIJI V1.46b plugins (ImageJ) were developed by the Imaging Core Facility (CIMA, Universidad de Navarra) for image analysis. Plugins contain image-processing operations, like colour segmentation, filtering and particle counting, which facilitate tissue segmentation from background and allow determination of the ratio of positive area versus total tissue area. Using FIJI plugins, collagen, MDR3 or γH2AX present in livers were quantified. The quantification was expressed as the percentage of positive tissue area (collagen), positive nuclei (γH2AX), or percentage of positive-stained area normalized to MDR2 signal observed in WT mice (MDR3).

**Statistical analysis**. Data are presented as mean values ± standard deviation and were statistically analysed separately for males and females using a one-way ANOVA test and Tukey's multiple comparison test or using unpaired $t$ test when there were only two groups ($p < 0.05$ was considered significant) with GraphPad Prism 7.05 software (GraphPad Software Inc., CA, USA).

**Reporting summary**. Further information on research design is available in the Nature Research Reporting Summary linked to this article.

## Data availability

The source data underlying Figs. 1c, 3a–c, 4a–c, 5a–c, 6a, b, and 7a–g, and Supplementary Figs. 1, 2, 3a, 4a, b, 5a, 6a, b, 7a, b, 8a, b, and 9a, b are provided as a Source Data file. The datasets generated during and/or analysed during the current study are available from the corresponding author on reasonable request.

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

## Acknowledgements

We thank Manuela Molina for excellent technical assistance, Dr. Carlos Rodríguez-Ortigosa and Sara Arcelus for analysis of serum biomarkers, and Dr. Matias A. Avila and Dr. María U. Latasa for help establishing the *Abcb4*−/− mouse colony and advice for preneoplastic marker analysis. We also thank Eneko Elizalde and all personnel at the Animal Facility, as well as the Morphology and Image Departments of Cima Universidad de Navarra.

## Author contributions

Study conception and design: N.D.W., G.G.-A., and C.S.; Acquisition of data: N.D.W., L.O., and J.M.-G.; Analysis and interpretation of data: N.D.W., L.O., and C.S.; Statistical analysis: N.D.W.; Drafting of the manuscript: N.D.W. and C.S. Important intellectual contributions during the study as well as critical revision of the manuscript: V.F., A.D., B.B., and G.G.-A.; Study supervision: N.D.W., C.S., and G.G.-A.

## Competing interests

Drs. Weber, Ferrer, Douar, Bénichou, and González-Aseguinolaza are all employees of Vivet Therapeutics, a startup biopharmaceutical company developing gene therapy approaches for treatment of rare metabolic diseases. All other authors declare no competing interests.
