## [Peer Review File · Nature Communications]

Reviewers' comments:

Reviewer #1 (Remarks to the Author):

This is an interesting study in which AAV8-mediated gene transfer of human ABCB4 gene into mouse liver has been used to effect improvement of phenotype in mice with progressive familial intrahepatic cholestasis (PFIC). There are a number of complexities to the mouse model, and these complicate interpretation of the study results. The authors should address the following:

1. Can the authors quantify the improvement in floppase activity for the codon-optimized vs. wt molecule? There needs to be some sort of benchmark for the co molecule vs wt. Similarly, is there any information on the physiologic function of the B and C isoforms? Given cytosolic localization, it is hard to understand what role they play in normal physiology.
2. An additional factor not discussed by the authors that may play a role in the sex difference in results is that the promoter used in these studies has been previously reported to perform better in males (publications by Nathwani and Nienhuis).
3. In the section of Results headed "AAV-MDR3-Aco treatment can partially restore PC levels in bile, preventing liver pathology", the authors state "...mediated by MDR3 expression in biliary canaliculi." This seems misleading. The gene is actually expressed in hepatocytes, and the gene product is secreted into the canaliculi, correct? This should be stated more clearly.
4. A puzzling result is the much lower level of genomes and transcripts in the 3E13 vg/kg injected mice compared to the 1E14. This is similar to the "hockey stick" dose response described by Pasi et al., (NEJM 2017) in a trial for hemophilia A. The reasons for this are completely unclear, and would seem to present some difficulty for scaling up into larger animals. Where is the inflection point? What causes it? These are both quite high doses, yet one seems to have hardly any effect. Some discussion of the dose-effect is warranted. Better yet would be some filling in of the dose-response curve, or injection of mice at an earlier stage, which might provide data before liver inflammation begins to confound results.
5. The results in the female mice, if the authors are correct, would seem to indicate that if disease is too advanced, the therapy may not be successful. The authors allude to data on mice treated for the first time at later time points. What implications does this have for therapeutic application?
6. The results would seem to suggest that the best course will be to treat all animals (or subjects) with two high doses, as this insures the best results on therapeutic outcome. Are there any disadvantages to these high doses? What happens if animals are followed for longer than 12 weeks?
7. In Fig 5, please state in the legend at what point the animals were sacrificed.
8. In the male mice shown in Figs. 5a and 5b, there seems to be a bimodal distribution of transcript levels (5b) despite tighter clustering of gene copy number. Is there an explanation for this? Can the authors comment?
9. Some references look dated or incorrect. For the trials in RPE65-mediated inherited retinal dystrophy, there is a published Phase 3 study (Lancet), and the regulatory bases of approval are also published online (through FDA and EMA). These are more appropriate references to the only licensed AAV product.

Reviewer #2 (Remarks to the Author):

This paper by Weber et al. describes the therapeutic effects of AAV human MDR3 vector treatment of Abcb4 KO mice. Treatment of Absb4 KO mice led to expression of human MDR3 in mice with increased biliary phospholipid secretion, reduced liver enzymes and liver fibrosis. From that authors conclude that MDR3 gene therapy may represent a suitable treatment option for PFIC3 patients. The paper is well written, the story is easy to follow, the quality of the figures is fine. However the novelty of the findings in regard to the used method on one hand and its potential therapeutic relevance is rather limited, since there are already numerous papers showing therapeutic effectiveness of several drugs and also human hepatocyte transplantation in Abcb4 KO mice. Consequently, the information that MDR3 transduction to young Abcb4 KO mice by a well known technique partially rescues their cholestatic phenotype represents not a major breakthrough.

1. This is proof of concept and the results show that MDR3 reintegration by viral delivery has

functionally worked.

2. Mice were only two weeks of age at the transduction. It would be additionally necessary to test this strategy in older animals with an already established phenotype, to learn if disease phenotypes are reversible.

3. Knowing what's happening with the tumors in transduced mice would significantly increase the impact of the paper.

Reviewer #3 (Remarks to the Author):

The study "Gene therapy of progressive familial intrahepatic cholestasis type 3 in a clinically relevant mouse model" by N. Weber and colleagues reports on the successful treatment of Abcb4-deficient mice by an adeno-viral vector-based gene transfer of human MDR3. This study is very well conducted and structured.

The study provides several interesting aspects of MDR3 and PFIC3-treatment: it addresses the differences of the three known isoforms A, B and C of MDR3. It provides evidence of different effectiveness of wild-type MDR3-A and codon-optimized MDR3-A. Importantly, it is shown that expression of human MDR3 in Mdr2-deficient mice improves the liver disease, that the threshold for treatment success is low (2-3 % of wt Mdr2 expression) and that repetition of vector-injection improves the effectiveness. The tight correlation of MDR3-expression and surrogate markers of disease is significant.

There are only minor points to be considered:

1. Details about the codon-optimization are missing. Is the sequence available?
2. It is not clear, why untreated wild-type mice were used as controls (instead of wt-mice treated with the empty vector). Likewise, Abcb4^{-/-} control mice could have been treated with the empty vector instead of saline-injection.
3. Figure 5A: How is the AAV-PCR-signal in saline-injected control-mice explained? Is this just a background signal? The same question arises concerning the quantification of MDR3-immunohistochemistry (Fig. 5C).
4. The differences in treatment between males and females is striking. Is there any role of the A1AT-promoter?
5. Figure-legends of figures 4 and 5: the meaning of the data "F(3,26)=23,41" etc. is unclear.
6. Page 11: it is stated, that 4000 μ M of PC in bile corresponds to 12-13% of PC-levels in normal mice. An exact reference for this statement should be given.
7. Figure 5C: MDR3-expression was measured by immunohistochemistry (IHC) and was normalized to the Mdr-expression in wt mice; however, wt mice express Mdr2 (the ortholog to human MDR3). What makes the authors sure, that the anti-ABCB4/MDR3 antibody has equal affinity for MDR3 and Mdr2?

Dear Reviewers,

The manuscript entitled ““Gene therapy correction of progressive familial intrahepatic cholestasis type 3 in a clinical relevant mouse model”” has been revised according to each of the Reviewers’ suggestions. Please find below a point-by-point reply to each of your comments (responses in red). All new data and changes suggested by you have also been highlighted in yellow in the manuscript.

Reviewer #1 (Remarks to the Author):

This is an interesting study in which AAV8-mediated gene transfer of human ABCB4 gene into mouse liver has been used to effect improvement of phenotype in mice with progressive familial intrahepatic cholestasis (PFIC). There are a number of complexities to the mouse model, and these complicate interpretation of the study results. The authors should address the following:

1. Can the authors quantify the improvement in floppase activity for the codon-optimized vs. wt molecule? There needs to be some sort of benchmark for the co molecule vs wt. Similarly, is there any information on the physiologic function of the B and C isoforms? Given cytosolic localization, it is hard to understand what role they play in normal physiology.

There appears to be no improvement in floppase activity for the co vs. wt MDR3. We carried out an activity assay *in vitro* comparing the six MDR3 variants and have appended the results to the manuscript (Figure 1c). Although there was no difference in activity between the co and wt versions of MDR3, there were significant differences between the three isoforms with isoform A being significantly more active than isoforms B or C. We hypothesize that indeed this reduced activity is because of their predominantly cytosolic localization. Although there is no evidence of a physiological role for isoforms B or C, cDNAs coding for these two variants were obtained from human liver, suggesting that they could be expressed *in vivo* (Van der Blik et al., EMBO J. 1987 6, 3325-31). In pursuant of the development of a gene therapy product to serve as a replacement for a mutated *ABCB4* gene, we chose to test all three predicted isoforms prior to selecting the most promising one. We added text to the Results on page 5 and to the Discussion on page 11:

2. An additional factor not discussed by the authors that may play a role in the sex difference in results is that the promoter used in these studies has been previously reported to perform better in males (publications by Nathwani and Nienhuis).

Although gender could play a role in promoter activity, we were unable to locate any published research pointing to sex differences in gene therapy efficiency related to the particular promoter that we used (alpha-1 anti-trypsin). Nathwani published research in 2003 (Davidoff et al., Blood 2003 Jul 15;102(2):480-8) showing a direct relationship of androgens in male mice with gene transfer in hepatocytes, regardless of the promoter, cDNA, or mouse strain. The reason for the difference is that AAV transduction efficiency, independently of AAV serotype or mouse strain, is always lower in females. These observations have also been corroborated by us and other groups (Pañeda et al., Hum Gene Ther. 2009 Aug;20(8):908-17, Berraondo et al., Hum Gene Ther. 2006 Jun;17(6):601-10). We have added these references as well as a comment to the Discussion on differences in transduction efficiency between males and females (page 12).

3. In the section of Results headed "AAV-MDR3-Aco treatment can partially restore PC levels in bile,

preventing liver pathology", the authors state "...mediated by MDR3 expression in biliary canaliculi." This seems misleading. The gene is actually expressed in hepatocytes, and the gene product is secreted into the canaliculi, correct? This should be stated more clearly.

Although this reviewer is correct that MDR3 is expressed in hepatocytes, it is not secreted into the canaliculi; rather, it localizes to the cell membrane on the canaliculi where it carries out the function of phosphatidylcholine transport across the membrane into the bile in the canaliculi. We have edited the sentence in Results as follows (page 7): "...mediated by MDR3 expression in hepatocytes and localization to the cell membrane of the biliary canaliculi."

4. A puzzling result is the much lower level of genomes and transcripts in the 3E13 vg/kg injected mice compared to the 1E14. This is similar to the "hockey stick" dose response described by Pasi et al., (NEJM 2017) in a trial for hemophilia A. The reasons for this are completely unclear, and would seem to present some difficulty for scaling up into larger animals. Where is the inflection point? What causes it? These are both quite high doses, yet one seems to have hardly any effect. Some discussion of the dose-effect is warranted. Better yet would be some filling in of the dose-response curve, or injection of mice at an earlier stage, which might provide data before liver inflammation begins to confound results.

We agree that an inflection point exists in the dose response curve, which we have further filled in by treating mice with three additional doses. These new results allowed us to determine the inflection point with more accuracy, being between 0.7 and 1x10E14 VG/kg. We believe that when the vector dose is not sufficient to achieve a certain threshold level of MDR3 expression, the disease progresses leading to a loss in AAV genomes and MDR3 transcripts, probably due to hepatocyte death and regeneration, which results in a dilution in the number of AAV genomes. In contrast, when the treatment achieves sufficient MDR3 expression, the disease is prevented and AAV genomes are not lost at the same rate. This would explain the "hockey stick" response that was observed. These new data have been added to the manuscript as Supplementary Figure 5, and have been included in the text of the Results section (page 9). Both the previous and new data were from mice injected at 2 weeks of age, prior to the appearance of liver inflammation.

5. The results in the female mice, if the authors are correct, would seem to indicate that if disease is too advanced, the therapy may not be successful. The authors allude to data on mice treated for the first time at later time points. What implications does this have for therapeutic application?

Our initial studies on mice treated at a higher age were performed with a reporter vector and with lower doses, in which we saw slightly decreased transduction in four-week-old compared to two-week-old animals. We have since returned to this line of questioning and have analyzed the therapeutic effect of mice treated one time with 1E14 VG/kg of AAV-MDR3-Aco at 5 weeks of age. We chose this age because five-week-old *Abcb4* KO mice already present symptoms of liver inflammation and fibrosis (Supplementary Figure 9). Interestingly, in this study we observed a similar outcome, in that all of the treated males and approximately half of females achieved a therapeutic effect, while the other half of the females showed a transient effect that waned after around 6 weeks following treatment (Figures 6-7 and Supplementary Figure 9). This result is very promising and points to the potential for translation to the clinic in which humans will be treated only after diagnosis of MDR3 deficiency implying the presence of symptoms. Results from this experiment have been added to the Results on page 10 and the Discussion on page 14.

6. The results would seem to suggest that the best course will be to treat all animals (or subjects) with

two high doses, as this insures the best results on therapeutic outcome. Are there any disadvantages to these high doses? What happens if animals are followed for longer than 12 weeks?

Indeed, treatment with two subsequent high doses achieved the best outcome in both males and females. Translating this course directly to the clinic may be challenging due to the generation of AAV-specific antibodies following the first treatment. In our study, the low age of the mice (and an immature immune system) at first treatment may in part be a reason why the second treatment was so successful. In human patients, it may be necessary to provide immunosuppressive agents at the time of treatment, utilize different serotypes and/or incorporate antibody-depleting therapy prior to subsequent treatments in order to achieve similar outcomes. In our study, twice-treated mice were followed for 16 weeks. In these mice the therapeutic effect was maintained up to the end of follow-up with no evidence of any adverse effects. We have revised the Discussion (page 14) as follows:

“We also report here an improvement in all parameters in mice treated twice with the AAV vector, which resulted in higher AAV transduction and MDR3 transgene expression compared to a single treatment, with no long-term adverse effects”.

7. In Fig 5, please state in the legend at what point the animals were sacrificed.

We have added the sacrifice timepoint to the legends for Figure 5 as follows: “Animals were sacrificed between 12 and 16 weeks after treatment.”

8. In the male mice shown in Figs. 5a and 5b, there seems to be a bimodal distribution of transcript levels (5b) despite tighter clustering of gene copy number. Is there an explanation for this? Can the authors comment?

One hypothesis for this bimodal distribution arose from the use of GAPDH transcripts as a normalizing gene in the MDR3 mRNA quantification. In more diseased animals (lower MDR3 expression), due to non-hepatocyte cell infiltrates, the amount of GAPDH mRNA will be increased from contributions from those cells while MDR3 mRNA will not be. To address this hypothesis, we reran the RT-qPCR using albumin as a normalizing gene (Figure 5b and Supplementary Figure 5b). This partly narrowed the gap in the bimodal distribution in once-treated males bringing it closer to the gap seen in the AAV genomes, but did not entirely eliminate it (vs. GAPDH: 20-fold increase between two subsets; vs. Alb: 8-fold increase; AAV genomes: 4.3-fold increase). A possible explanation for this result could be that in all mice a certain amount of AAV genomes become transcriptionally inactive. These genomes could constitute a larger fraction of the total AAV genome pool in mice treated with a lower dose of vector, resulting in proportionally lower levels of MDR3 mRNA. It may also be that these mice are more diseased due to having expression levels below the therapeutic threshold, which could possibly play a factor on AAV transcriptional activity.

9. Some references look dated or incorrect. For the trials in RPE65-mediated inherited retinal dystrophy, there is a published Phase 3 study (Lancet), and the regulatory bases of approval are also published online (through FDA and EMA). These are more appropriate references to the only licensed AAV product

As suggested by the reviewer we have updated the abovementioned references.

Reviewer #2 (Remarks to the Author):

This paper by Weber et al. describes the therapeutic effects of AAV human MDR3 vector treatment of *Abcb4* KO mice. Treatment of *Abcb4* KO mice led to expression of human MDR3 in mice with increased biliary phospholipid secretion, reduced liver enzymes and liver fibrosis. From that authors conclude that MDR3 gene therapy may represent a suitable treatment option for PFIC3 patients. The paper is well written, the story is easy to follow, the quality of the figures is fine. However the novelty of the findings in regard to the used method on one hand and its potential therapeutic relevance is rather limited, since there are already numerous papers showing therapeutic effectiveness of several drugs and also human hepatocyte transplantation in *Abcb4* KO mice. Consequently, the information that MDR3 transduction to young *Abcb4* KO mice by a well known technique partially rescues their cholestatic phenotype represents not a major breakthrough.

As we detailed in the Introduction, UDCA therapy is the current standard of care for PFIC3 patients. However, it is only effective in roughly 50% of patients. Orthotopic liver transplantation is the only remaining option to treat individuals with the severest form of this disease. This medical outcome carries high associated costs and risks, as well as limited availability which leads to premature death. If an effective alternative option existed, a very high unmet medical need could be addressed. Our studies on the proposed AAV-mediated therapy have shown promising proof of concept results that point to further development of the product. We think that, as indicated in the Discussion section, one of the main novelties of our study is that we were able to correct the PFIC3 phenotype with a rather low percentage of hepatocyte vector transduction, which could allow for a mimic of gene therapies for diseases that involve secreted proteins, such as those for haemophilia A and B.

Human hepatocyte transplantation in *Abcb4*^{-/-} mice, although representing a very novel proof of concept approach, possesses many of the same hurdles inherent to whole organ transplant. In addition, this technology still needs to be improved for clinical use due to low cell engraftment and lack of long-lasting effects (Lansante et al., *Pediatr Res*, 2018. 83(1-2):232-240).

1. This is proof of concept and the results show that MDR3 reintegration by viral delivery has functionally worked.
2. Mice were only two weeks of age at the transduction. It would be additionally necessary to test this strategy in older animals with an already established phenotype, to learn if disease phenotypes are reversible.

As described above (response to reviewer 1, point 5), we have added to the manuscript results from KO mice treated at five weeks of age with one single dose of AAV-MDR3-Aco. The results of this study showed that the AAV treatment can overcome already established phenotype and reduce the disease biomarkers much in the same way as treatment in two-week-old mice (Figures 6-7 and Supplementary Figure 9).

3. Knowing what's happening with the tumors in transduced mice would significantly increase the impact of the paper.

The *Abcb4*^{-/-} mouse (FVB.*Mdr2*^{-/-}) has been reported to develop HCC starting at 12 months of age, and it takes 16 months for all mice to develop tumours (Potikha et al. *Hepatology* 2013, 58:1, 192-204). Although the question of whether our AAV treatment can prevent tumour formation in the livers of mice is highly interesting, this outcome is rarely observed in human PFIC3 patients (Davitt-Spraul A, et al. *Semin Liv Dis*, 2010. 30(2):134-46; and Wendum D, et al., *Virchows Arch*, 2012. 460(3):291-8). Due to

the delayed tumour appearance in the FVB background strain mice, our studies were not sufficiently long to obtain direct data on the appearance of tumours, and since our primary objective is a preclinical proof of concept, we did not prioritize measuring this disease outcome. Furthermore, the mice suffer from a mortality rate of 10-20% at 8 months of age independent of treatment (data not shown), and carrying out our study to a terminal timepoint of over 12 months would have added unnecessary risks of losing study participants and reducing the power to detect significant differences in the other disease parameters.

That being said, we have carried out an analysis of several markers of preneoplasia in our study population, which were sacrificed between 3.5 and 4.5 months of age. We evaluated expression levels of glycine N-methyltransferase (GNMT) and alpha fetoprotein (AFP) by qRT-PCR, which are downregulated and upregulated, respectively, in both human HCC and *Abcb4* KO liver tumours (Katzenellenbogen et al. Mol Cancer res. 2007; Territo et al., BMC Medical Imaging, 2015; Zhou et al. J. Hepatology 2017). In addition, we also quantified phosphorylated histone H2AX (γ H2AX) by immunohistochemistry, an indicator of DNA double strand breaks that has been reported to be a good preneoplastic marker of HCC in *Abcb4* KO mice (Barash et al. PNAS 2010). Our results showed greater evidence of preneoplasia in saline-treated KO males than in saline-treated KO females compared to WT. In the males that responded to our AAV treatment (achieving bile PC concentrations above 3900 μ M), there was a significant improvement in γ H2AX, compared to either saline-treated or non-responder males, and a trend towards improvement in GNMT and AFP. For treated females, significant improvements were observed in AFP and γ H2AX. However, there was no significant difference between saline-treated and WT females for any of the markers, despite trends in AFP and γ H2AX.

Our new results, which point to a likely reduction in tumour formation in mice that responded to our therapy have been added to the manuscript (page 8 and Supplementary Figures 3 and 4).

Reviewer #3 (Remarks to the Author):

The study "Gene therapy of progressive familial intrahepatic cholestasis type 3 in a clinically relevant mouse model" by N. Weber and colleagues reports on the successful treatment of *Abcb4*-deficient mice by an adeno-viral vector-based gene transfer of human MDR3. This study is very well conducted and structured.

The study provides several interesting aspects of MDR3 and PFIC3-treatment: it addresses the differences of the three known isoforms A, B and C of MDR3. It provides evidence of different effectiveness of wild-type MDR3-A and codon-optimized MDR3-A. Importantly, it is shown that expression of human MDR3 in *Mdr2*-deficient mice improves the liver disease, that the threshold for treatment success is low (2 -3 % of wt *Mdr2* expression) and that repetition of vector-injection improves the effectiveness. The tight correlation of MDR3-expression and surrogate markers of disease is significant.

There are only minor points to be considered:

1. Details about the codon-optimization are missing. Is the sequence available?

Due to legal reasons and the need to protect Vivet Therapeutics' intellectual property, we are not able to disclose the codon optimized sequence at this time.

2. It is not clear, why untreated wild-type mice were used as controls (instead of wt-mice treated with

the empty vector). Likewise, *Abcb4*^{-/-} control mice could have been treated with the empty vector instead of saline-injection.

We agree with the reviewer that using *Abcb4*^{-/-} mice treated with an empty AAV vector or with an AAV vector expressing a reporter gene could have constituted an additional valuable control. However, we believe that the best control in these studies are saline-treated KO mice in which the disease followed a natural course that could be directly compared with the effects in mice treated with the therapeutic vector. An empty vector could lead to an immune response in the animals that might increase inflammation, potentially exacerbating disease symptoms. Furthermore, in gene therapy clinical trials empty vectors are not used for the placebo arm.

Untreated wild type mice were included to provide values of the biological parameters for healthy mice. Again, treating them with empty vector would run the risk of not producing healthy values.

3. Figure 5A: How is the AAV-PCR-signal in saline-injected control-mice explained? Is this just a background signal? The same question arises concerning the quantification of MDR3-immunohistochemistry (Fig. 5C).

Yes, this is just background signal. The saline-treated mice Ct values for amplification of the AAV genome are similar to the Ct values for no template controls in the qPCR reactions. Since these samples do harbor copies of the GAPDH gene, the value of this background signal can be calculated and shown on the graph in the form of copies relative to genomic GAPDH copies ($2^{\Delta Ct}$), while no template controls cannot.

For MDR3 expression as detected by IHC, the program for identifying and quantifying brown area on the slide produces a background reading on samples with no MDR3 expression, which are graphed as the saline-treated group. An example of an image for MDR3 IHC of a saline-treated KO animal is shown in Figure 1D, lower right.

4. The differences in treatment between males and females is striking. Is there any role of the A1AT-promoter?

We believe that differences in therapeutic effect between males and females are due to increased liver transduction efficiency in males as a result of androgen-mediated stabilization of viral genomes (see point 2 in responses to Reviewer 1). This effect was observed to be independent of the promoter used, and we do not believe that the A1AT promoter induces different gene regulation depending on sex.

5. Figure-legends of figures 4 and 5: the meaning of the data "F(3,26)=23,41" etc. is unclear.

These are the F values and degrees of freedom (numerator and denominator) for the ANOVA statistical tests. We have changed all the figure legends to contain an explanation for the degrees of freedom (numerator and denominator).

We have rechecked our statistical analysis and have decided that one-way ANOVA test is more appropriate so as to compare males only with males and females only with females.

6. Page 11: it is stated, that 4000 μ M of PC in bile corresponds to 12-13% of PC-levels in normal mice. An exact reference for this statement should be given.

This calculation stems from an analysis of our own data, which showed WT mice to have $31120 \pm 4530 \mu$ M (males) and $33666 \pm 2295 \mu$ M (females). We have changed this sentence (now in page 13):

“The threshold of 4000 μ M of bile PC concentration represents about 12-13% the levels in our WT mice, which is comparable to the amount of *Abcb4*^{+/+} hepatocyte repopulation reported by de Vree et al. (Gastroenterology 2000; 119, 1720-1730).”

7. Figure 5C: MDR3-expression was measured by immunohistochemistry (IHC) and was normalized to the Mdr-expression in wt mice; however, wt mice express Mdr2 (the ortholog to human MDR3). What makes the authors sure, that the anti-ABCB4/MDR3 antibody has equal affinity for MDR3 and Mdr2?

The epitope region used for creating the α -MDR3 antibody used is 98% homologous between human and mouse (49/50 homology in amino acid sequence), and the one change is a functional equivalence change according to protein blast (288E>K) (<https://www.lsbio.com/antibodies/abcb4-antibody-mdr3-antibody-aa252-301-ihc-wb-western-ihc-plus-ls-b5729/140335>). The producer tested and found this antibody positive for reactivity against human, monkey, mouse, rat, bovine, dog, hamster, and rabbit. Although this information does not guarantee 100% equal affinity for MDR3 and Mdr2, we utilized WT mouse Mdr2 expression as a reference point in which to calculate and compare the amount of expression in KO mice for the different treatments and at different sacrifice timepoints. Our conclusions do not hinge upon the precision of the value of 2-3% of WT levels (although we believe this value to be accurate), but rather on the existence of a threshold expression level only above which a therapeutic effect can be achieved.

REVIEWERS' COMMENTS:

Reviewer #1 (Remarks to the Author):

The authors have responded appropriately to all points raised in the review. The reviewer is still unclear as to the details of how the hepatocyte-synthesized MDR3 makes its way to the membrane of the bile canaliculi. Is MDR3 found throughout the hepatocyte membrane? Is there a localization signal that directs it to the portion of the membrane that forms the bile canaliculi? This aspect of liver histology is not clear to me and may be unclear to others as well.

Reviewer #2 (Remarks to the Author):

no further comments

Reviewer #3 (Remarks to the Author):

Dear authors,
the manuscript has been further improved and all questions have been addressed.

One point remains open (question 2, review 3): the optimal use of controls. Your own answer specifies the problem: the vector might induce some inflammation - this effect may also apply to the ko-mice, which were treated with the AAV-MDR3-Aco vector. Inflammatory responses induce cholestasis, which may have an impact of MDR3-expression in gene-transfected ko-animals. Therefore this point needs to be considered when further research is done with this model.

Dear Reviewers,

The manuscript entitled *“Gene therapy correction of progressive familial intrahepatic cholestasis type 3 in a clinical relevant mouse model”* has been revised according to each of the Reviewers' suggestions. Please find below a point-by-point reply to each of your comments (responses in red).

Reviewer #1 (Remarks to the Author):

The authors have responded appropriately to all points raised in the review. The reviewer is still unclear as to the details of how the hepatocyte-synthesized MDR3 makes its way to the membrane of the bile canaliculi. Is MDR3 found throughout the hepatocyte membrane? Is there a localization signal that directs it to the portion of the membrane that forms the bile canaliculi? This aspect of liver histology is not clear to me and may be unclear to others as well.

Previous studies performed by other groups have addressed the mechanism by which MDR3 and other transporters expressed by hepatocytes, like BSEP, are specifically transported to their canalicular membrane and are not present in other areas of the membrane in these cells^{1,2}.

In particular, MDR3 is first synthesized in the endoplasmic reticulum and is then transported to the plasma membrane via the Golgi apparatus. In contrast to most transmembrane proteins, which first traffic to the basolateral membrane and then undergo transcytosis to reach the apical plasma membrane, ABC transporters, especially MDR3, BSEP and MDR1, are directly targeted to the membrane of bile canaliculi². The molecular mechanism of this transport is still poorly understood, but it seems to involve particular membrane microdomains³. It has also been described that the sub-apical compartment (SAC), also known as the common endosome (CE), can function as an internal reservoir of these proteins in order to adapt their final levels at the plasma membrane

Since we have expressed an MDR3 protein containing exactly the same amino acid sequence as the endogenous protein, we believe that our recombinant protein will follow the same transport route to the canalicular membrane that takes place in normal hepatocytes. We felt that an in-depth explanation of the biology surrounding this is extraneous to the findings included in our paper.

References:

1. H. Kipp, I.M. Arias. Newly synthesized canalicular ABC transporters are directly targeted from the Golgi to the hepatocyte apical domain in rat liver. *J Biol Chem*, 275 (2000), pp. 15917-15925
2. Falguières T, Aït-Slimane T, Housset C, Maurice M. ABCB4: Insights from pathobiology into therapy. *Clin Res Hepatol Gastroenterol*. 2014 Oct;38(5):557-63
3. T. Aït-Slimane, G. Trugnan, I.S.C. Van, D. Hoekstra. Raft-mediated trafficking of apical resident proteins occurs in both direct and transcytotic pathways in polarized hepatic cells: role of distinct lipid microdomains. *Mol Biol Cell*, 14 (2003), pp. 611-624

Reviewer #2 (Remarks to the Author):

no further comments

Reviewer #3 (Remarks to the Author):

One point remains open (question 2, review 3): the optimal use of controls. Your own answer specifies the problem: the vector might induce some inflammation - this effect may also apply to the ko-mice, which were treated with the AAV-MDR3-Aco vector. Inflammatory responses induce cholestasis, which may have an impact of MDR3-expression in gene-transfected ko-animals.

Therefore this point needs to be considered when further research is done with this model.

We thank the reviewer for this interesting observation and will take it into consideration when doing further research with this animal model.

Saline-treated KO mice are needed to gauge the extent of the disease without treatment. If our therapeutic-vector-treated mice had shown evidence of disease such as cholestasis or liver inflammation that was treatment-dependent, a control of an empty vector would make sense to elucidate if the outcome was a result of the AAV vector or of the AAV vector plus the transgene payload. But this wasn't the case. Since our AAV-MDR3-treated mice improved in the disease symptoms (and the saline-treated mice exhibited the symptoms), we think this control was unnecessary.